**Data Availability Statement:** Data are available in manuscript.

# Pharmacological inhibition of carnitine palmitoyl transferase 1 inhibits and reverses experimental autoimmune encephalitis in rodents

**Anne Skøttrup Mørkholt**[1], **Michal Krystian Oklinski**[1], **Agnete Larsen**[2], **Robert Bockermann**[3], **Shohreh Issazadeh-Navikas**[3], **Jette Goller Kloth Nieland**[4], **Tae-Hwan Kwon**[5], **Angelique Corthals**[6], **Søren Nielsen**[1,4], **John Dirk Vestergaard Nieland**[1] *

**1** Department of Health Science and Technology, Aalborg University, Aalborg, Denmark, **2** Department of Biomedicine, Aarhus University, Aarhus C, Denmark, **3** Biotech Research and Innovation Centre, Copenhagen University, Copenhagen N, Denmark, **4** Meta-IQ, ApS, Aarhus C, Denmark, **5** Department of Biochemistry and Cell Biology, School of Medicine, Kyungpook National University, Taegu, Korea, **6** Department of Science, John Jay College of Criminal Justice, City University of New York, New York, New York, United States of America

* jdn@hst.aau.dk

## Abstract

Multiple sclerosis (MS) is a neurodegenerative disease characterized by demyelination and inflammation. Dysregulated lipid metabolism and mitochondrial dysfunction are hypothesized to play a key role in MS. Carnitine Palmitoyl Transferase 1 (CPT1) is a rate-limiting enzyme for beta-oxidation of fatty acids in mitochondria. The therapeutic effect of pharmacological CPT1 inhibition with etomoxir was investigated in rodent models of myelin oligodendrocyte glycoprotein- and myelin basic protein-induced experimental autoimmune encephalitis (EAE). Mice receiving etomoxir showed lower clinical score compared to placebo, however this was not significant. Rats receiving etomoxir revealed significantly lower clinical score and lower body weight compared to placebo group. When comparing etomoxir with interferon-β (IFN-β), IFN-β had no significant therapeutic effects, whereas etomoxir treatment starting at day 1 and 5 significantly improved the clinical scores compared to the IFN-β and the placebo group. Immunohistochemistry and image assessments of brain sections from rats with EAE showed higher myelination intensity and decreased expression of CPT1A in etomoxir-treated rats compared to placebo group. Moreover, etomoxir mediated increased interleukin-4 production and decreased interleukin-17α production in activated T cells. In conclusion, CPT1 is a key protein in the pathogenesis of EAE and MS and a crucial therapeutic target for the treatment.

## Introduction

One of the key functions of lipids in the CNS is to build and maintain the myelin sheath on the axons of neurons, as well as to facilitate protein transfer from oligodendrocytes to the myelin

**Funding:** A grant from Lundbeckfonden, R191-2015-1118, financed the studies. J.G.K.N. and S.N. disclose being employed and have financial interest in Meta-IQ. The funders provided support in the form of the drug Etomoxir used in the animal study as well as know-how on the drug, but did not have any additional role in the study design, data collection and analysis, decision to publish, or preparation of the manuscript. The specific roles of these authors are articulated in the 'author contributions' section.

**Competing interests:** J.G.K.N. and S.N. declares conflict of interest as owner of Meta-IQ ApS, who provided the etomoxir for the study. This does not alter our adherence to PLOS ONE policies on sharing data and materials.

sheaths [1,2]. It is well established that metabolism of lipids is significantly altered in multiple sclerosis (MS) patients [3–7]. Previous studies that examined the lipid levels of MS, bipolar disorder or schizophrenia patients demonstrated a significant reduction in the mono-unsaturated and saturated fatty acids lipid concentrations, specifically of 16:1 (palmitate) and 18:1 lipids (oleate lipids) [8–10]. Although polyunsaturated fatty acid levels were not examined in these studies, the shift in lipid levels was likely due to changes in metabolic pathways, such as switch in the balance between the metabolism of glucose and of lipids. This metabolic redirection, when occurring in the brain, leads to the reduced levels of poly- and mono-unsaturated and saturated fatty acids in the myelin sheath and cerebral spinal fluid. Moreover, cholesterol levels in the body are changed and correlate with MS progression [5,6]. Evidence supporting this pathogenesis is shown by positron emission tomography analyses of MS lesions which reveal a greatly reduced glucose metabolism [7,11].

In addition to lower levels of constitutional lipids, an upregulation of enzymes involved in lipid metabolism has been identified in MS lesions. One of the most significant enzymes in the metabolism of lipids is carnitine palmitoyl transferase 1A (CPT1A), which is abundantly expressed in the brain and most organs where it catalyzes the rate-limiting step in beta-oxidation. CPT1 is localized at the outer mitochondrial membrane, where it couples an acyl-carnitine group with lipids, and facilitates transport across the outer mitochondrial membrane. After transport of acyl-carnitine, CPT2 localized at the inner mitochondrial membrane removes the carnitine group from the lipid molecule which enables its degradation via beta-oxidation. Biological systems switch their metabolism from glucose to lipid, i.e. enhanced catabolism of lipids, under locally occurring hypoxic stress. This happens in order to prevent overproduction of lactate and avoid the associated toxicity. Increased catabolism of lipids under situations of stress induces local production of prostaglandin E2 (PGE2) [12], thereby attracting and activating the immune system. Moreover, prolonged lipid catabolism activated under hypoxia can result in mitochondrial dysfunction.

In the MS lesions, *CPT1A* expression is greatly increased [13], which correlates with a decrease in lipid concentration in the myelin sheath due to an increased beta-oxidation. Loss of lipids in the myelin sheath results in a functional impairment, manifested by increasing signaling time and an escalation in energy expenditure. In addition, lipids provide a protective function for the proteins expressed in the myelin sheath. If the lipids of the myelin sheath are no longer functional or present at all, proteins in myelin such as myelin basic protein (MBP) have some arginine's converted to citrulline by peptidyl arginine deiminase, thereby making them immunogenic [14–17]. Moreover, under hypoxic conditions lipid metabolism is associated with increased prostaglandin production [12,18], and enhanced chemotaxis of immune cells to the exposed myelin sheath proteins.

Based on these mechanisms, we therefore hypothesize that blockage of CPT1A function may result in both therapeutic and preventative effects in the progression of MS. To examine this hypothesis, we studied the efficacy of the CPT1 blocker etomoxir for the treatment of advanced experimental autoimmune encephalitis (EAE) in rodents [19]. The changes of MBP, CPT1A and ferritin expression were examined in the EAE rat model. Moreover, the effect of etomoxir on inflammation and production of cytokines were examined.

## Materials and methods

### Animals

All animal experiments were conducted according to NIH guidelines and were approved by the Danish National Committee for Ethics in Animal Experimentation (2007/561-1364 and 2015-15-0201-00647). Six-week-old female C57BL/6 mice (n = 42) were bred and kept at

conventional animal facilities at the University of Copenhagen. Two-month-old female Lewis rats (n = 42 and n = 55) were bred and kept at conventional animal facilities at the University of Aarhus. All animals were maintained under standardized conditions of light and temperature, with a 12-h day/night cycle and food and water ad libitum. During the progression of EAE leading to motor disabilities, animals had water in a petri dish and soaked chow for easy intake to ensure sufficient liquid and nutrients intake.

## Experimental autoimmune encephalomyelitis immunization

C57BL/6 mice were immunized subcutaneously (s.c.) in the flank with 200 μg of MOG$_{35-55}$ peptide emulsified in complete Freund´s adjuvant (CFA) containing 0.1 mg *Mycobacterium tuberculosis* (Becton Dickinson) along with an intraperitoneal injection of 200 ng pertussis toxin (List Biological Laboratories Inc.) on the day of immunization and two days later. In Lewis rats, EAE was induced by intradermal injection at the base of the tail with an emulsion consisting of 100 μg MBP from guinea pig (Sigma-Aldrich) suspended in CFA with the addition of 0.2 mg of *Mycobacterium Tuberculosis* (Becton Dickinson). The animals were monitored daily, weighed and clinically scored according to Table 1. No animals were permitted to lose more than 20% body weight (compared to the starting point of EAE immunization) and to go beyond score 4 [20].

## Drug administration

At indicated time points (Fig 1), animals were treated daily s.c. either with 1 mg/kg etomoxir (Meta-IQ ApS) diluted in non-sterile olive oil at 37 ˚C or 200,000 IU IFN-β (Extavia, Aalborg University Hospital) every other day. The placebo groups received daily injections with saline or olive oil (non-sterile). Etomoxir is a potent inhibitor of CPT1, which blocks mitochondrial fatty acid beta-oxidation and exerts its effects on both the adaptive and innate immune response as well as mobilizing anti-inflammatory effects [21,22].

## Immunohistochemical and immunofluorescent staining

Rats were anesthetized with isoflurane and intracardiac perfusion with 0.01M PBS was followed by fixation with 4% PFA/PBS (pH 7.4). The brains were isolated and placed for storage in 4% PFA/PBS at 4 ˚C. Brains were cut into 2 mm thick coronal sections and subjected for washing procedure in cold running tap water for 4 h and in PBS for following five days at 4 ˚C with daily exchange of fresh PBS. Sections were dehydrated in ethanol, cleared overnight in xylene and embedded in paraffin. The paraffin-embedded tissue blocks were cut into 2 μm thick sections on a rotary microtome (Leica Microsystems), dewaxed, rehydrated and underwent either peroxidase or fluorescent immunolabeling. Sections were stained for MBP mouse

**Table 1. Clinical scoring of animals.**

| Disease score | Clinical symptoms of EAE |
|:---:|:---|
| 0 | No clinical signs of EAE |
| 1 | Limp tail |
| 2 | Paresis of one or two hind limbs |
| 3 | Unilateral hind leg paralysis |
| 4 | Bilateral hind leg paralysis |
| 5 | Bilateral hind leg paralysis and incontinence or moribund |

Clinical assessment of animals induced with experimental autoimmune encephalomyelitis (EAE).

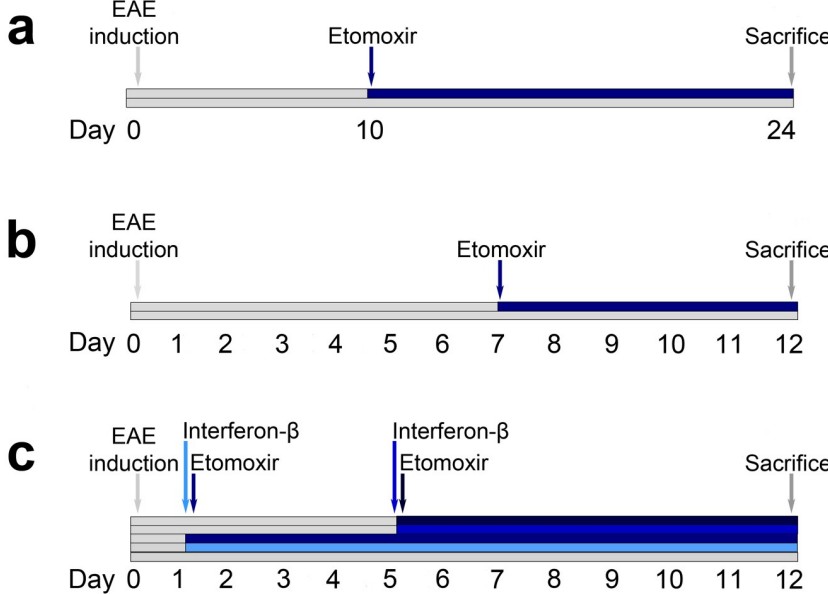

**Fig 1. Timeline of the experimental autoimmune encephalomyelitis (EAE) models in mice and rats.** Mice were injected with myelin oligodendrocyte glycoprotein (MOG)$_{35-55}$ and at day 10–12 animals started showing EAE symptoms. Therefore, treatment with etomoxir (1 mg/kg/day s.c., n = 21) or placebo (n = 21) was initiated at day 10 (a). Rats were injected with myelin basic protein (MBP) and daily treatment of etomoxir (1 mg/kg/day s.c., n = 16) or placebo (n = 26) was started at day 7 (b). Rats were injected with MBP and treated with etomoxir (1 mg/kg/day s.c.) at day 1 (n = 10) or day 5 (n = 10), interferon-β (IFN-β) (200,000 IU, every other day, s.c.) day 1 (n = 10) or day 5 (n = 10), or placebo (n = 10) onwards after induction of EAE (c). All studies were terminated when animals showed a disease score of > 4 or > 20% weight loss.

antibody (1:100, ab62631, Abcam) in connection with goat anti-mouse (1:200, P0447, Dako) horseradish peroxidase-conjugated secondary antibody and ferritin rabbit antibody (1:500, ab81444, Abcam) in connection with anti-rabbit antibody (1:200, P0448, Dako) horseradish peroxidase-conjugated secondary antibody accordingly to the procedure described previously [23]. Bright field microscopy was carried out using DM5500B microscope (Leica). Immunofluorescence labeling of CPT1A was performed with CPT1A primary mouse antibody (1:100, ab128568, Abcam) and anti-mouse Alexa fluor 488 secondary antibody (1:200, A21202, Thermo Fisher). TO-PRO⁻3 Iodide (1:1000, T3605, Thermo Fisher) was used as the nuclear counterstain. Whereas, MBP quantification was performed with antibody mentioned earlier in connection with anti-mouse Alexa fluor 555 secondary antibody (1:200, A-21422, Thermo Fisher). Laser scanning on confocal microscopy was carried out (Leica DMI6000CS, Wetzlar, Germany).

## Intensity quantification of immunofluorescent labeling

Images designated for fluorescent labeling intensity quantification were taken under the resolution of 1024 x 1024 pixels. Total of 12 images (six from brainstem and six from cerebellum) from coronal sections at approximately Bergma-13, in possibly similar locations, were acquired from each etomoxir-treated animal (n = 3) and placebo-treated animal (n = 3). All other microscopy settings, such as laser power, gain, offset, contrast, gamma values intensity and pinhole size were kept identical during the whole acquisition process. Single channel gray scale images were used to assess the integrated density value of CPT1A- and MBP-labeled

fluorescent sections in ImageJ software after setting a constant threshold level covering desired labeling.

## Intracellular cytokine analysis

Experiments involving human blood samples were carried out in accordance with the approved guidelines and regulations according to the Declaration of Helsinki. Informed consent forms of these healthy donors have been obtained. The study and use of human blood material was approved by Ethical Committee for Region North Denmark (N-20150073). HPBL were isolated, cultured and stained in accordance with Mørkholt et al. [21]. Cells were stained with FITC mouse anti-human IFN-γ (561057, BD Biosciences), IL-8 (340509, BD Biosciences) or TNF-α (562082, BD Biosciences), and PE mouse anti-human IL-4 (562046, BD Biosciences) or IL-17α antibodies (560438, BD Bioscience). Cells were analyzed by a FACS flow cytometer (Beckmann Coulter).

## Applied statistics

All statistics were performed using Graph Pad Prism software. Unpaired t tests and ordinary one-way ANOVA with Tukey´s multiple comparisons post hoc test were used to analyze clinical EAE parameters. Chi-square and Fisher´s exact tests were used to analyze differences in number of animals showing normal behavioral clinical scores. Two-way ANOVA with either Tukey´s or Sidak´s multiple comparisons post hoc tests was used to compare clinical score and weight data over time between the different treatment groups, as well as to compare flow cytometric expression of cytokines in treated and untreated cells. Expression of CPT1A and MBP was evaluated by an unpaired t test. Acquired data are presented as mean ± SEM. P values of 0.05 were considered statistically significant.

## Results

### Blockage of CPT1 has therapeutic effects in rodent models of EAE

*CPT1A* is upregulated in the brain lesions of MS patients [13]. As CPT1 is a rate-limiting step in mitochondrial lipid metabolism of fatty acid beta-oxidation, we tested if blocking of CPT1 with etomoxir (selectively blockage of CPT1 including CPT1A) is effective in the treatment of EAE progression by reversing functional deficits. EAE was induced by immunization of C57BL/6 mice (n = 42) and Lewis rats (n = 42 and n = 55) with the myelin oligodendrocyte glycoprotein (MOG)$_{35-55}$ peptide and MBP respectively, as described in methods. Ten to twelve days after immunization, the C57BL/6 mice showed signs of movement disturbances of the tail and/or hind legs. At this time point (day 10), mice were treated with etomoxir in non-sterile olive oil or with the placebo (pure non-sterile olive oil) (Fig 1a).

The mice were daily tested for body weight changes and clinical disease scores. After two weeks of treatment (day 24), the animal study was terminated and the remaining mice were sacrificed. Five days after the first injection (day 15), the overall mean clinical scores improved in the etomoxir-treated mice (Fig 2a). There were no significant differences in body weight between place and etomoxir-treated mice (Fig 2b). The classical EAE parameters showed that mice treated with etomoxir revealed lower mean maximum EAE score, later mean day of disease onset and lower disease incidence compared to the placebo group (Fig 2c). Etomoxir treatment showed a therapeutic effect as 47.6% of mice had normal behavioral scores at day 24 compared to 28.6% in the placebo group (Fig 2c).

The therapeutic efficacy of the treatment was also tested in Lewis rat EAE model using MBP. Normally, disease onset is seen at day 8 to 9 after disease induction and progresses

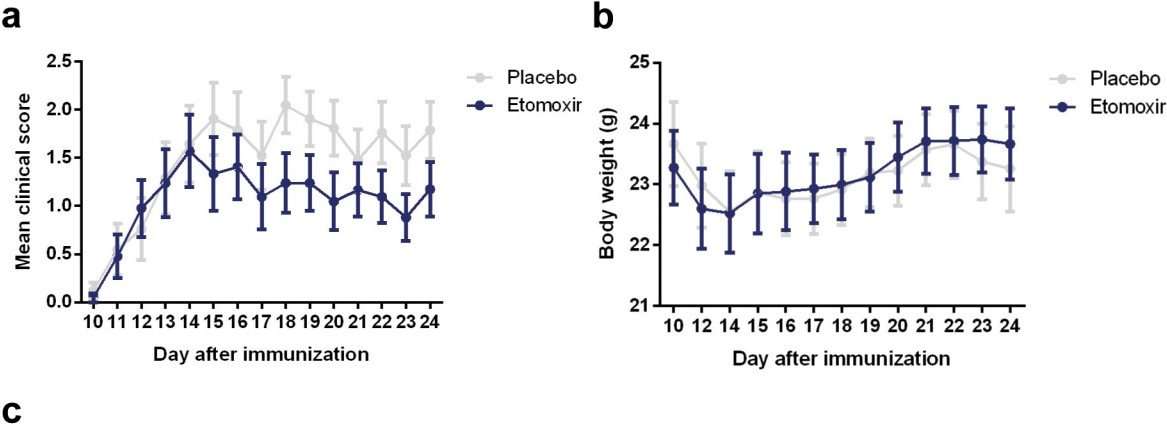

**Fig 2. Blockage of carnitine palmitoyl transferase 1A (CPT1A) by etomoxir in a mice experimental autoimmune encephalomyelitis (EAE) model.** Two weeks of treatment with etomoxir (1 mg/kg/day s.c., n = 21) compared to placebo (n = 21) treatment of mice with myelin oligodendrocyte glycoprotein (MOG)$_{35-55}$-induced EAE. From day 15, etomoxir treatment revealed lower clinical score compared to animals receiving placebo (a). The average body weight was higher in the mice subjected to etomoxir treatment as compared to placebo-treated animals (b). Statistically significant differences between etomoxir and placebo mice were not observed for maximum disease score, disease onset, incidence (c). Data are represented as mean ± SEM.

rapidly. Therefore, treatment of the rats with etomoxir or placebo was started at day 7 (Fig 1b) [24]. Moreover, rats receiving etomoxir exhibited significantly lower disease scores at day 11 compared to the rats receiving placebo (p = 0.0013; two-way ANOVA with Sidak's post test) (Fig 3a). Placebo-treated rats showed significantly decreased body weight compared to the etomoxir-treated rats (day 9 p = 0.0259; day 10 p = 0.0029; day 11 p = 0.0006; two-way ANOVA with Sidak's post test) (Fig 3b). Additionally, etomoxir-treated rats showed significant lower mean maximum EAE score (p = 0.0236; unpaired t test) and later mean day of onset (p = 0.0081; upaired t test) compared to the placebo group (Fig 3c). At day 11, none of the placebo-treated rats showed a normal behavioral score and 73% had already been sacrificed due to a disease score of 4. In contrast, 25% of the etomoxir-treated rats had normal behavioral scores, which were highly significant compared to the placebo group (p = 0.0074; chi-square) (Fig 3c).

## Superior therapeutic effects produced by CPT1 inhibition compared to IFN-β in a rodent model of EAE

CPT1 blockage by etomoxir was compared to interferon-β (IFN-β), as this is one of the current first-line treatments of MS. Since it has been shown that IFN-β exhibits clinical efficacy in a rat EAE model when given early, we compared the two pharmacological treatment regimens when given at day 1 or day 5 after induction of disease (Fig 1c) [25]. This model produced very severe outcomes, since most of the placebo and IFN-β-treated rats were sacrificed at day 11 due to high disease scores. IFN-β treatment started at day 1 and 5 did not produce therapeutic

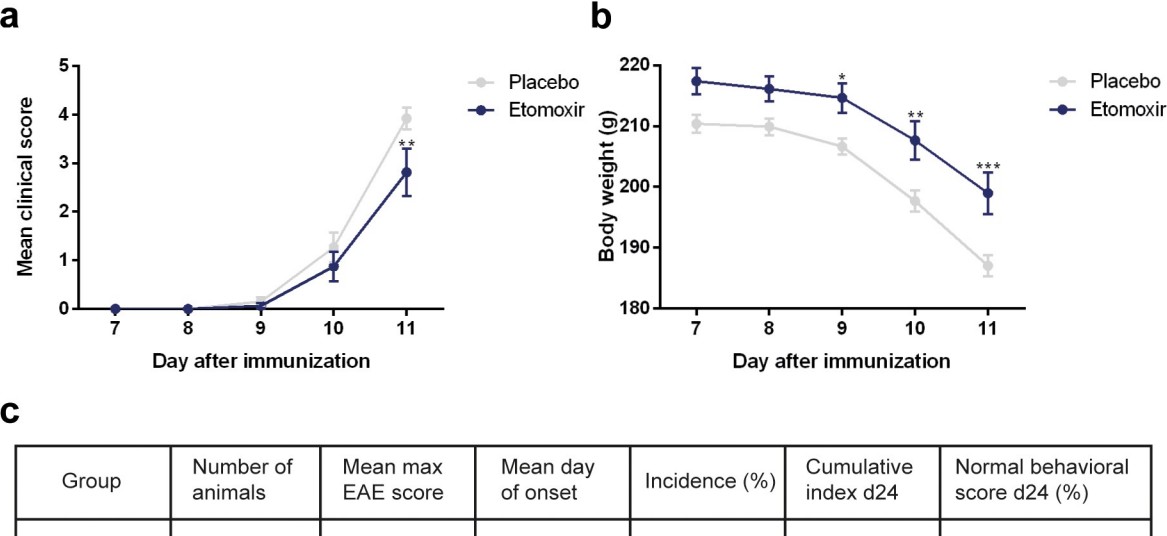

**Fig 3. Blockage of carnitine palmitoyl transferase 1A (CPT1A) by etomoxir in a rat experimental autoimmune encephalomyelitis (EAE) model.** Rats were subjected to myelin basic protein (MBP)-induced EAE and treated either with etomoxir (1 mg/kg/day s.c., n = 16.) or placebo (n = 26). Etomoxir-treated rats exhibited significantly lower disease scores at day 11 compared to rats receiving placebo analyzed by a two-way ANOVA with Sidak's multiple comparisons post hoc tests (a). The results from the two-way ANOVA with Sidak's multiple comparisons post hoc test showed significantly higher body weight in the rats subjected to etomoxir treatment as compared to placebo-treated animals at day 9, 10 and 11 (b). Statistically significant differences between etomoxir and placebo treated rats were observed for maximum disease score and disease onset (unpaired t test) (c). Etomoxir treatment resulted in 25% of the rats showing normal behavioral scores compared to 0% after placebo treatment analyzed by a Fisher´s exact test (c). Data are represented as mean ± SEM. Number of asterisks indicates the level of statistical significance ($^{*}$p < 0.05, $^{**}$p < 0.01, $^{***}$p < 0.001).

effects, which was the same in the placebo-treated group. In contrast, etomoxir treatment demonstrated therapeutic effects at day 10 when initiated at day 1 compared to placebo, IFN-β at day 1 and day 5 (placebo p = 0.0347; IFN-β at day 1 p<0.0001; IFN-β at day 5 p = 0.0004; two-way ANOVA with Sidak's post test) (Fig 4a), and when started at day 5 compared to IFN-β at day 1 and day 5 (IFN-β at day 1 p = 0.0004; IFN-β at day 5 p = 0.0046; two-way ANOVA with Sidak's post test) (Fig 4a). Additionally, at the last day rats treated with etomoxir at day 1 exhibited lower disease scores compared to animals receiving placebo (p = 0.0133), IFN-β at day 1 (p = 0.0004) and IFN-β at day 5 (p = 0.0133) (Fig 4a). Rats started treatment with etomoxir at day 5 showed significantly lower disease score compared to IFN-β at day 1 (p = 0.0133; two-way ANOVA with Sidak's post test) (Fig 4a). Etomoxir, however, did not significantly affect the body weight over time (Fig 4b). Moreover, etomoxir treatment initiated at day 1 revealed a significant lower mean maximum EAE score compared to IFN-β at day 1 (p = 0.0102; one-way ANOVA with Tukey´s post test) (Fig 4c).

## Etomoxir decreases CPT1A expression and improves myelination and ferritin deposits

Immunohistochemistry provided additional evidence for the therapeutic effect of etomoxir treatment compared to the placebo group in the rat EAE model. Etomoxir-treated rats had

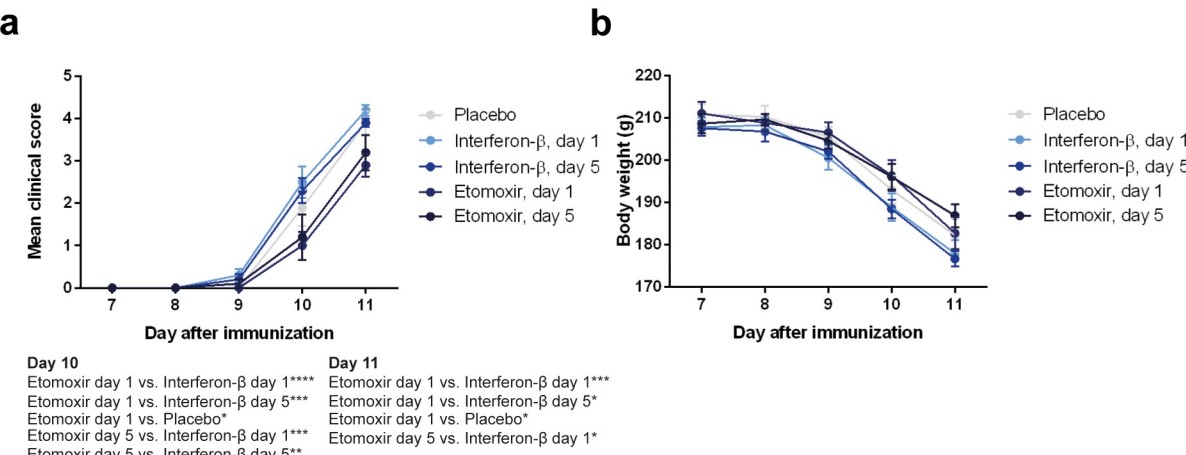

**Day 10**
Etomoxir day 1 vs. Interferon-β day 1****
Etomoxir day 1 vs. Interferon-β day 5***
Etomoxir day 1 vs. Placebo*
Etomoxir day 5 vs. Interferon-β day 1***
Etomoxir day 5 vs. Interferon-β day 5**

**Day 11**
Etomoxir day 1 vs. Interferon-β day 1***
Etomoxir day 1 vs. Interferon-β day 5*
Etomoxir day 1 vs. Placebo*
Etomoxir day 5 vs. Interferon-β day 1*

c

| Group | Number of animals | Mean max EAE score | Mean day of onset | Incidence (%) | Cumulative index d24 | Normal behavioral score d24 (%) |
|---|---|---|---|---|---|---|
| Placebo | 10 | 3.90 ± 0.28 | 10.20 ± 0.13 | 8/10 (80.0) | 39 | 0/10 (0.0) |
| Interferon-β day 1 | 10 | 4.20 ± 0.13 | 9.80 ± 2.00 | 9/10 (90.0) | 42 | 0/10 (0.0) |
| Interferon-β day 5 | 10 | 3.90 ± 0.10 | 9.90 ± 0.18 | 9/10 (90.0) | 39 | 0/10 (0.0) |
| Etomoxir day 1 | 10 | 2.90 ± 0.28* | 10.50 ± 1.17 | 5/10 (50.0) | 29 | 0/10 (0.0) |
| Etomoxir day 5 | 10 | 3.20 ± 0.42 | 10.50 ± 0.22 | 4/10 (40.0) | 32 | 0/10 (0.0) |

**Fig 4. Comparing the effects of etomoxir and interferon-β (IFN-β) in a rat experimental autoimmune encephalomyelitis (EAE) model.**
Rats were subjected to myelin basic protein (MBP)-induced EAE and treated either with etomoxir (1 mg/kg/day s.c.) at day 1 (n = 10) or day 5 (n = 10), IFN-β (200,000 IU s.c.) every other day from day 1 (n = 10) or day 5 (n = 10), or placebo (n = 10) onwards after EAE induction. Results from the RM two-way ANOVA with Tukey's multiple comparisons post hoc test showed that rats receiving etomoxir at day 1 and day 5 exhibited significantly lower disease scores compared to IFN-β day 1, IFN-β day 5 and placebo at day 10 and 11 (a). The average body weight was higher in etomoxir-treated rats compared to IFN-β-treated animals, though this was not significant (b). Statistically significant difference between etomoxir and IFN-β both initiated at day 1 was observed for maximum disease score (unpaired t test) (c). Data are represented as mean ± SEM. Number of asterisks indicates the level of statistical significance (*p < 0.05, **p < 0.01, ***p < 0.001, ****p < 0.0001).

markedly increased intensity of MBP labeling with no evident pathological lesions (Fig 5a–5d), particularly pronounced in the cerebellum (Fig 5b–5d) when compared with placebo-treated rats (Fig 5e–5h). In the cerebellum of etomoxir-treated rats, MBP labeling in the white matter was more intense compared with placebo-receiving rats (Fig 5b and 5f). Similarly, in the etomoxir group, more intense MBP labeling was noted in granular and molecular cerebellar layers in the form of fibrous bundles protruding from the white matter (Fig 5c and 5d). Whereas, in the placebo group, characteristic MBP-labeled fibrous bundles were sparse and labeling pattern was more dotted with barely noticeable threadlike structure (Fig 5g and 5h). In the brainstem, differences in MBP labeling intensity were less evident than that of cerebellum. Nevertheless, in the etomoxir-treated group, MBP-labeled fibrous structures laterally

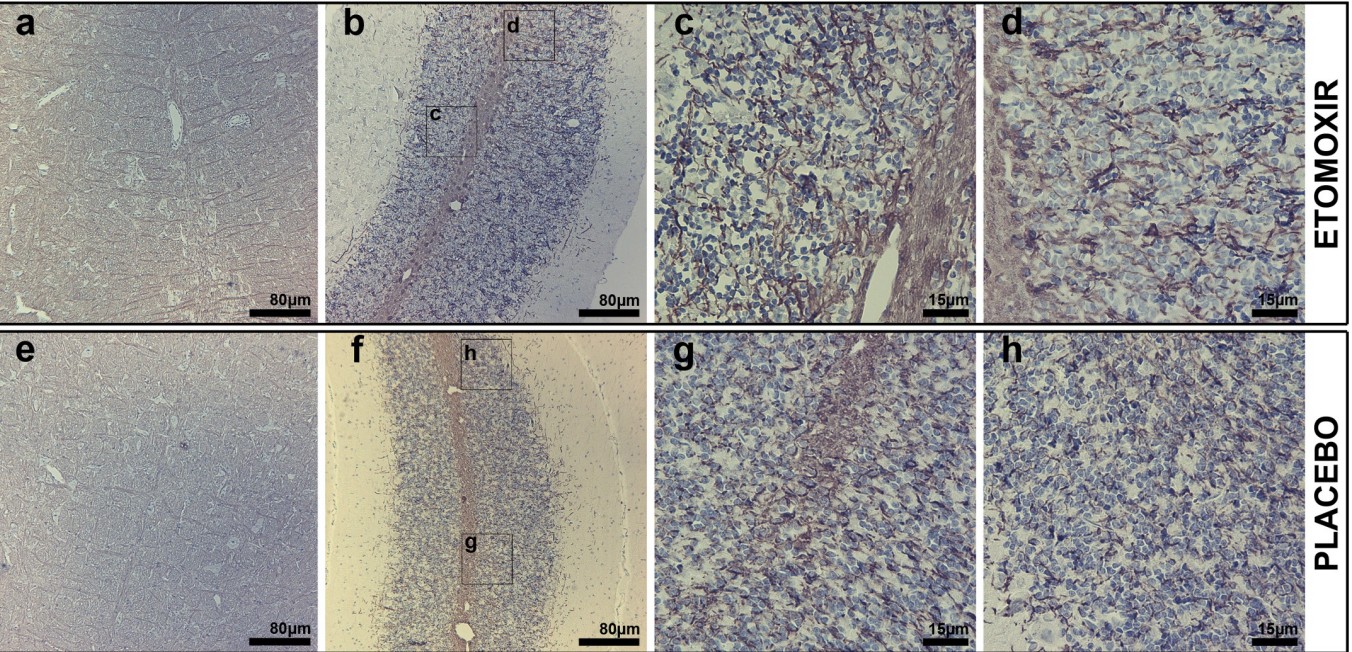

**Fig 5. Immunoperoxidase microscopy of myelin basic protein (MBP) in coronal brainstem and cerebellum of etomoxir-treated rats (a-d) and placebo rats (e-h).** The etomoxir-treated group shown to possess higher MBP-labeling in the brainstem visualized as fibrous structures laterally protruding from *nucleus raphe magnus* and *medial longitudinal fascicle* compared to the group receiving placebo (a and e). Likewise, in the cerebellar white matter of etomoxir-treated rats, MBP-labeling was markedly more intense compared with placebo-receiving rats (b and f). MBP axons ensheathing in etomoxir-treated rats was labeled in the form of fibrous bundles in the cerebellar white matter, granular and molecular cerebellar layers, whereas in placebo animals MBP-labeled fibrous bundles were sparse and labeling pattern was more (c,d,g and h).

protruding from the *nucleus raphe magnus* and the *medial longitudinal fascicle* (Fig 5a) were labeled more intensely than in the placebo group (Fig 5e).

Immunofluorescent signal intensity of CPT1A in paraffin-embedded brain sections from etomoxir-treated rats was significantly lower in both brainstem (p = 0.0083; unpaired t-test) and cerebellum (p = 0.0237; unpaired t test) compared to the placebo group (Fig 6a–6c), confirming the downregulation of CPT1 in etomoxir-treated group. These data showed that blocking CPT1, which specifically blocks the metabolism of lipids, produced therapeutic

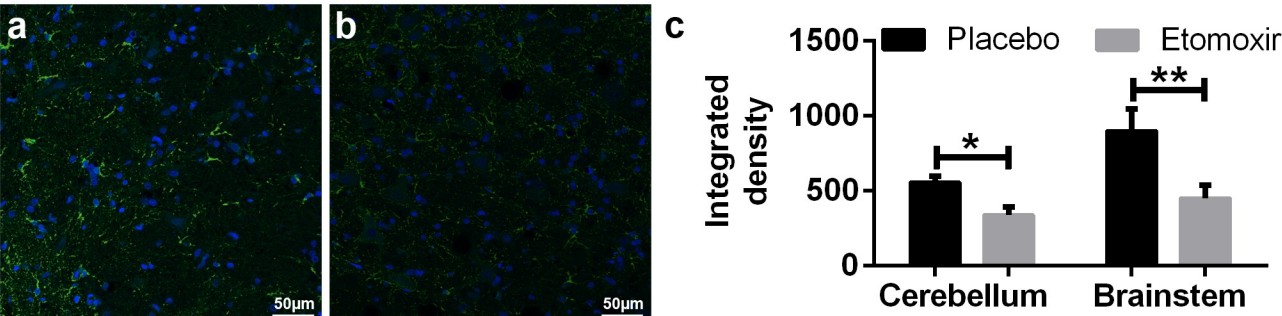

**Fig 6.** Immunofluorescent images of carnitine palmitoyl transferase 1A (CPT1A) in placebo-receiving rats (n = 3) (a) and etomoxir-treated rats (n = 3) (b). CPT1A signal intensity was significantly lower in both cerebellum and brainstem of the etomoxir-treated animals compared to the placebo-treated animals illustrating the blocking effect of etomoxir treatment (c). CPT1A was labeled with green and TO-PRO-3 was used as nuclei counterstain overlaid with blue color. Data are represented as mean ± SEM. Number of asterisks indicates the level of statistical significance (*p < 0.05, **p < 0.01).

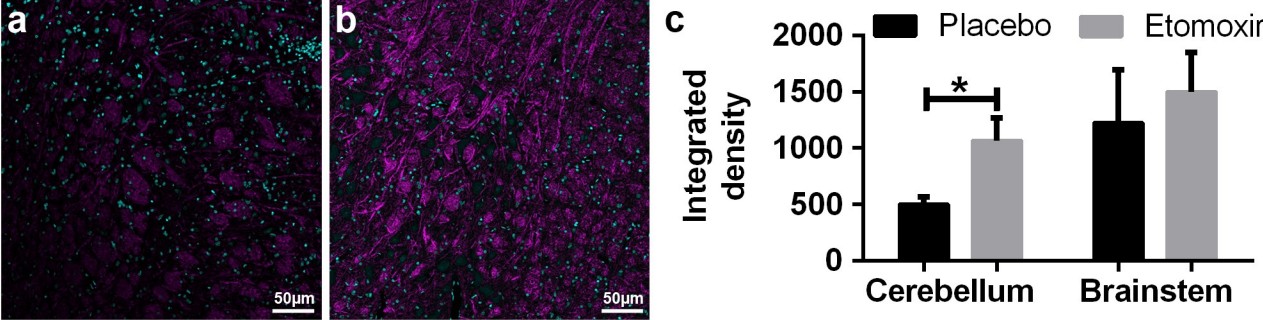

**Fig 7.** Immunofluorescent images of myelin basic protein (MBP)-labeling in the brainstem of placebo-receiving rats (n = 3) (a) and etomoxir-treated rats (n = 3) (b). MPB-labeling was significantly higher in the cerebellum of etomoxir-treated rats and represented similar tendency in the brainstem, though this was not statistical significantly changed (c). MBP was visualized with Alexa fluor 555 antibody and overlaid with magenta color, whereas nuclear counterstain TO-PRO-3 was overlaid with cyan. Data are represented as mean ± SEM. Number of asterisks indicates the level of statistical significance (*p < 0.05).

effects in the animal models of MS. Consequently, fluorescent MBP quantification have confirmed qualitative peroxidase-labeling (Fig 7a and 7b) and showed significantly higher level of MBP-labeling in the cerebellum (p = 0.0192; unpaired t test) with matching trend in the brainstem (Fig 7c).

Rats receiving etomoxir showed reduced ferritin labeling in the brainstem and cerebellum when compared to the placebo group (Fig 8a–8d). In the cerebellum and brainstem of etomoxir-treated rats, ferritin labeling was noticed mostly around blood vessels (Fig 8a and 8c), whereas in the placebo group labeling was readably more intense and noticed also around and inside the cells (presumably neurons) (Fig 8b and 8d).

Additionally, MBP fluorescent labeling together with TO-PRO-3 nuclei stain was used to illustrate immune cell infiltration. Among sections from etomoxir-treated rats cell aggregations with readably decreased MBP-labeling surrounding them had not been encountered (Fig 9a), whereas in the sections from placebo-receiving rats described cellular aggregates that were encountered especially in the brainstem (Fig 9b and 9c).

### Decreased IFN-γ and IL-17α expression in activated T cells after etomoxir treatment

In MS, the pro-inflammatory immune response is associated with disease pathogenesis [26–28], therefore the effect of etomoxir on the inflammatory response was also investigated. Human peripheral blood lymphocytes were stimulated with staphylococcus enterotoxin B (SEB) for 48 h in the presence or absence of 100 μM etomoxir. After incubation, the cytokine expression was measured by intracellular flow cytometry. Etomoxir blocked the production of interferon-γ (IFN-γ) (58%), interleukin-8 (IL-8) (11%), interleukin-17α (IL-17α) (64%) and the tumor necrosis factor-α (TNF-α) (55%). A small increase of interleukin-4 (IL-4) (27%) production was observed. The T cell response was directed from a Th1, Th17α response to a Th2 type T cell response. Etomoxir also mediated significantly increase in IL-4 production and decrease in IL-17α production (IL-4 p = 0.0065; IL-17α p = 0.0102; two-way ANOVA with Tukey's post test) (Fig 10).

### Discussion

For a long time MS has been regarded as an autoimmune disease, where the immune system has been proposed as the primary cause of the disease. Most treatment strategies have been

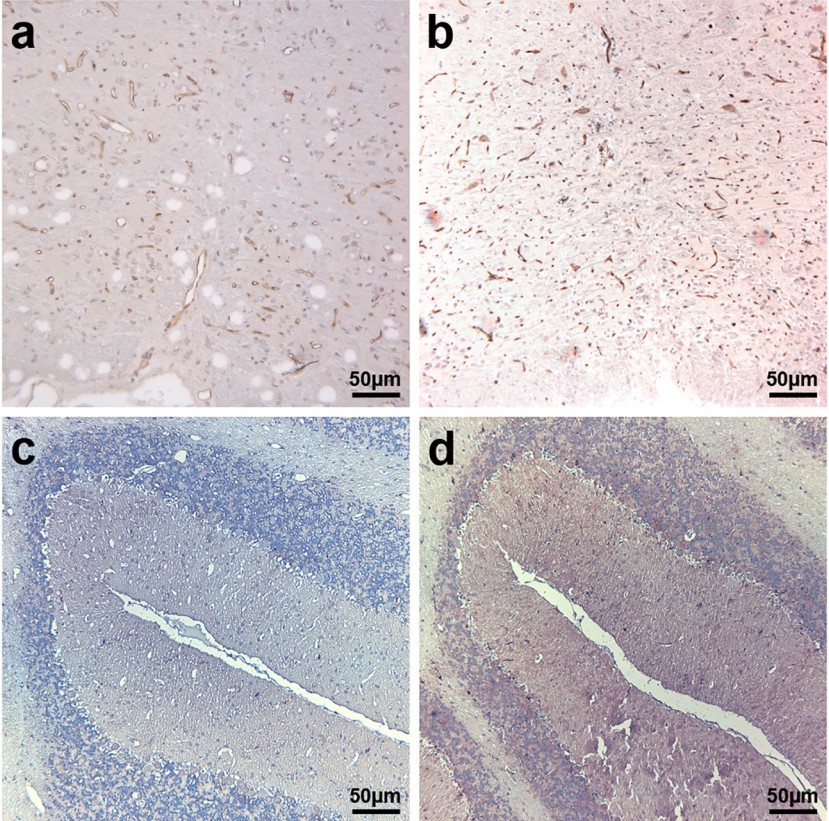

**Fig 8. Immunoperoxidase staining of ferritin in cerebellum and brainstem from etomoxir-treated rats and placebo-receiving rats.** Rats receiving etomoxir showed reduced ferritin labeling in both brainstem (a) and cerebellum (c) when compared to the placebo group, respectively (b and d). In the placebo group vastly more intense labeling was noticed around blood vessels and noticed also around and inside the cells (presumably neurons) (b and d).

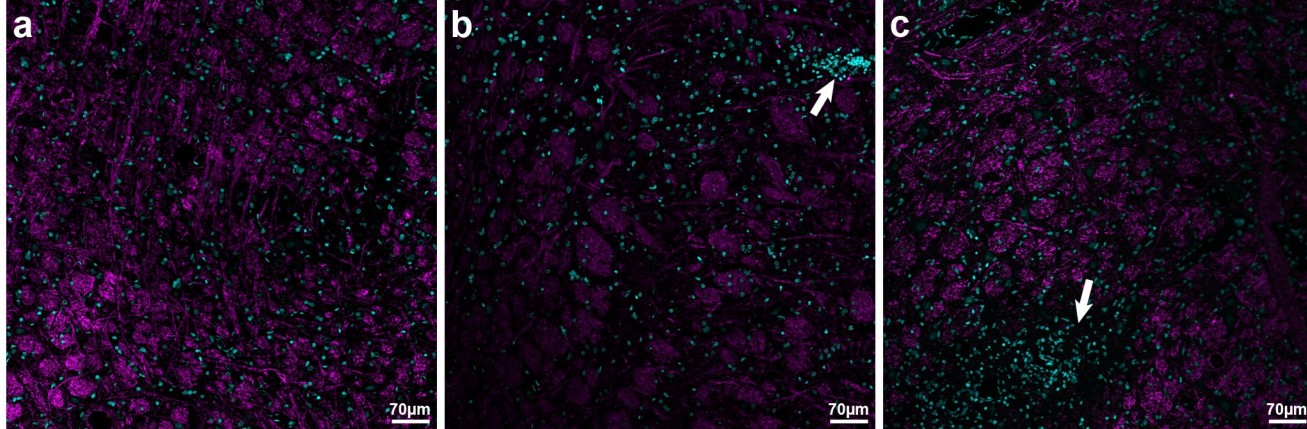

**Fig 9. Myelin basic protein (MBP) fluorescent labeling together with TO-PRO-3 nuclei stain illustrating immune cell infiltration in etomoxir-treated rats (n = 3) and placebo-receiving rats (n = 3).** In the sections from placebo-receiving rats, TO-PRO-3-labeled cellular aggregates were encountered especially in the brainstem (indicated by arrows in b and c) and surrounded with decreased level of MBP-labeling. There were very spars cell aggregations in the sections from etomoxir-treated rats (a). MBP was visualized with Alexa fluor 555 antibody and overlaid with magenta color, whereas nuclear counterstain TO-PRO-3 was overlaid with cyan.

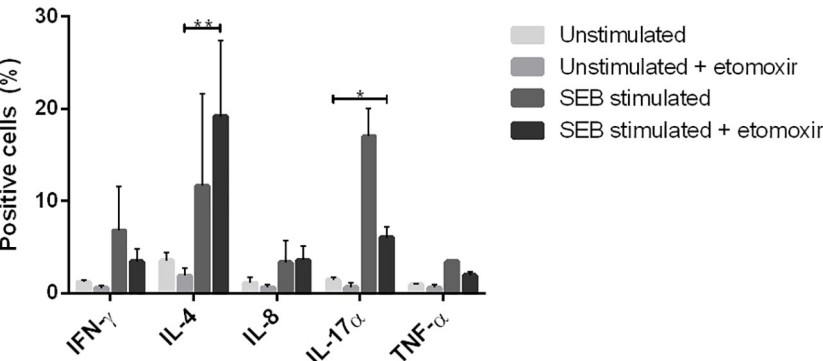

**Fig 10. Intracellular cytokine analysis of etomoxir-treated Staphylococcal enterotoxin B (SEB)-stimulated human peripheral blood lymphocytes.** Etomoxir revealed the lower percentage of positive cells for interferon-γ (IFN-γ), tumor necrosis factor-α (TNF-α) and interleukin-17α (IL-17α), but induced increased percent positive cells for interleukin-8 (IL-8) and interleukin (IL-4). Results were analyzed by a two-way ANOVA with a Tukey multiple comparison post hoc tests. Data are represented as mean ± SEM. Number of asterisks indicates the level of statistical significance (*p < 0.05, **p < 0.01).

focused solely on the modulation of the inflammatory component of the disease. Thus far, these treatments have been successful in suppressing or delaying relapsing episodes in relapsing-remitting MS and primary-relapsing MS. However, these interventions have not brought an obvious therapeutic effect on disease progression, particularly for the treatment of primary-progressive MS and secondary-progressive MS.

Consistent with the results demonstrated by Lieury *et al.* [13], we have found that *CPT1A* is upregulated in MS and EAE lesions. Compared to the normal healthy brain, the mean *CPT1A* expression is 179% in active MS plaque, 140% in chronic active plaque and 124% in chronic plaque (NCBI Geo Profiles). Moreover, the key role of CPT1A in MS is supported by studies regarding *CPT1A* mutations in humans. Two types of loss-of-function mutation in *CPT1A*, which inhibits the activity of the enzyme, have been identified in humans (Hutterite and Inuit populations) [29–31]. Common for both populations is highly reduced MS frequency compared to the Canadian population [32,33]. A study with mice having the CPT1A mutation mimicking the Inuit mutation is supporting this key role of CPT1a in the development of EAE, since these mice were resistant to EAE development [34].

Approximately 70% of brain tissue is comprised of lipids and thereby lipids are one of the most important components of the brain. Lipids constitute the myelin sheath, which is important for efficient transmission of signals through neurons with high speed and low energy. The half-life of these myelin lipids is approximately three days [35]. This replacement occurs efficiently in healthy brain tissues through maintenance of high lipid levels by the oligodendrocytes. In MS, however, *CPT1A* is upregulated, leading to lipid catabolism through accelerated beta-oxidation, as described by Lieury *et al.* [13]. This finding was further confirmed by evidence of decreased lipid levels in association with disease progression [3,36–38]. The loss of lipids in myelin exposes a citrullinated MBP (deamination of arginine to citrulline), which, in turns, increases the inflammatory and immune responses (both B and T cells) due to PGE2 production [14–17,39]. Thus, the neurites are slowly stripped from the protective myelin sheath. This inflammatory response in EAE can be blocked by etomoxir [40].

With conditions of stress (physical, pathological, psychological or immunological) follows a state of hypoxia in the brain, which causes the metabolism to shift locally from a glucose-based to a lipid-based metabolism. This will have an effect on the balance in the peroxisome

proliferator-activated receptor (PPAR) network [3]. Additionally, due to favored lipid metabolism the insulin receptor will become insensitive, furthering the lipid metabolism [41]. This process will particularly have an effect on the CNS as lipids play an important role in signal propagation and minimizing energy expenditure. The hypothalamic-pituitary-adrenal (HPA) axis will be hyper-activated and stimulating an immune response at the site of lipid metabolism, resulting in an immune attack on the myelin sheath proteins and oligodendrocytes [42,43]. Therefore, by blocking CPT1 with etomoxir, the inflammatory response will be downregulated and the lipid metabolism will be blocked resulting in restoration of glucose metabolism, increased insulin sensitivity, and restoration of the PPAR balance and HPA axis.

We hypothesized that the optimal treatment regimen for MS would be to block the runaway lipid catabolism, thereby maintaining or increasing lipid levels in the brain and brainstem. More lipids can then be used by oligodendrocytes to restore the altered lipidation of the myelin sheaths, restoring neuronal functions. As lipids are important in both building the myelin sheath and the activation of memory B and T cells, the inhibition of lipid catabolism blocks memory and inflammatory immune responses, leaving the regulatory immune response intact.

We demonstrated that inhibition of lipid catabolism by blocking CPT1 in animal models of EAE reverses disease progression (Figs 2, 3 and 4). In addition, comparison of CPT1A labeling fluorescent signal intensity in brain sections of EAE-induced rats resulted in significantly lower signal intensity in the etomoxir-treated rats versus the placebo group, thus confirming downregulated CPT1 levels after etomoxir treatment (Fig 6). Significant downregulation of CPT1A had occurred together with the markedly higher intensity of MPB labeling in cerebellum and brainstem (Fig 7). Consequently, improved MPB labeling in the cerebellum of etomoxir-treated rats (Fig 5) corresponded well with their lower clinical scores. Inhibition of CPT1 demonstrated effect on the ferritin levels, visualized by reduction in staining in the etomoxir-treated animals compared to the placebo group (Fig 8). Ferritin signals iron deposits which are a result of mitochondrial dysfunction [44,45]. The decreased brain metabolism together with dysfunctional mitochondria promote an environment for enhances oxidative stress. A study has demonstrated that the severity of EAE correlates with the gene expression of genes involved in oxidative stress, such as HO-1 and NOX2 [34]. Moreover, blocking lipid metabolism by etomoxir decreased the Th1 and Th17 inflammatory responses, which have been shown to be involved in the MS development [26–28]. Similarly, our results have shown much lower occurrence of cellular aggregates presumably belonging to immune infiltrates, encircled with increased MBP levels in the brainstem of etomoxir-treated rats (Fig 9). After two weeks of treatment in mice suffering from EAE, 47.6% of the mice had normal behavioral scores (Fig 2c) and in the rat EAE model, 25% of the etomoxir-treated animals had normal behavioral scores when treatment was started at day 7 (Fig 3c). The etomoxir-treated rats exhibited statistically significant lower clinical scores and body weight over time compared to the placebo group (Fig 3a and 3b). Furthermore, etomoxir-treated rats showed significantly later disease onset ($13.69 \pm 1.54$) compared to the placebo group ($10.31 \pm 0.13$). Comparing the treatment efficacy between etomoxir and IFN-β (the first line of medicine given currently to MS patients) showed that IFN-β treatment was ineffective at 1 or 5 days after induction of the disease in animals with EAE, whereas etomoxir had very pronounced efficacy by significantly lowering the clinical scores of the animals compared to animals receiving IFN-β day 1 and 5, and placebo (Fig 4a). Moreover, the maximum EAE score for rats treated with etomoxir initiated day 1 were significantly reduced compared to IFN-β day 1 (Fig 4c).

The overall conclusion from the presented data is that suppression of the enhanced lipid catabolism and beta-oxidation through CPT1A inhibition could provide therapeutic effects against MS. This may be assumed since the EAE model is a state-of-art *in vivo* animal model of

MS. It shows similarities to MS by both clinical and pathological features, and reveal the complex interaction of demyelination, inflammation, axonal loss and gliosis. On the other hand, there are some limitations by using EAE to investigate MS relations such as viruses (e.g. Epstein-Barr virus), psychological and social factors since these cannot be modelled properly in animal models. However, many of the drugs available for MS today have been developed and validated in studies using the EAE model [19].

In conclusion, the proposed metabolic switch from glucose to lipid could provide an explanation to the pathogenesis of MS, based on the data from our EAE experiments where blocking the metabolism of lipids stops disease progression. When the lipidation of the myelin sheath is defective, the blockage of an inflammatory response by common disease modifying drugs such as IFN-β can only slow down the disease progression. Thus our experiments with etomoxir not only provide a new avenue of treatment, but also provide new evidence for a paradigm shift when thinking about the pathophysiology of MS. Our data show that MS should be considered as a systemic disease, where altered lipid metabolism in the myelin sheath plays a primary role, and the immune system is the aggravating factor. Moreover, this approach will provide a new insight into therapeutic approaches for other neurodegenerative diseases such as Alzheimer´s disease and Parkinson´s disease, as well as Amyotrophic lateral sclerosis.

## Acknowledgments

The authors thank Meta-IQ for providing etomoxir, and animal caretakers at Aarhus University and Copenhagen University Animal Facility.

## Author Contributions

**Conceptualization:** Jette Goller Kloth Nieland, Søren Nielsen, John Dirk Vestergaard Nieland.

**Data curation:** Anne Skøttrup Mørkholt, Michal Krystian Oklinski.

**Formal analysis:** Anne Skøttrup Mørkholt, Michal Krystian Oklinski.

**Investigation:** Anne Skøttrup Mørkholt, Michal Krystian Oklinski, Agnete Larsen, Robert Bockermann, Shohreh Issazadeh-Navikas.

**Methodology:** Anne Skøttrup Mørkholt, Michal Krystian Oklinski, Agnete Larsen, Robert Bockermann, Shohreh Issazadeh-Navikas.

**Project administration:** Anne Skøttrup Mørkholt.

**Resources:** Jette Goller Kloth Nieland.

**Software:** Anne Skøttrup Mørkholt.

**Supervision:** John Dirk Vestergaard Nieland.

**Writing – original draft:** Anne Skøttrup Mørkholt.

**Writing – review & editing:** Anne Skøttrup Mørkholt, Michal Krystian Oklinski, Tae-Hwan Kwon, Angelique Corthals, Søren Nielsen, John Dirk Vestergaard Nieland.

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
