## [Decision Letter · Decision Letter 0]

25 Mar 2020

PONE-D-19-35522

Pharmacological inhibition of carnitine palmitoyl transferase 1 inhibits and reverses experimental autoimmune encephalitis in rodents

PLOS ONE

Dear Mr Nieland,

Thank you for submitting your manuscript to PLOS ONE. After careful consideration, we feel that it has merit but does not fully meet PLOS ONE’s publication criteria as it currently stands. Therefore, we invite you to submit a revised version of the manuscript that addresses the points raised during the review process.

We would appreciate receiving your revised manuscript by May 09 2020 11:59PM. To enhance the reproducibility of your results, we recommend that if applicable you deposit your laboratory protocols in protocols.io, where a protocol can be assigned its own identifier (DOI) such that it can be cited independently in the future. For instructions see: http://journals.plos.org/plosone/s/submission-guidelines#loc-laboratory-protocols

We look forward to receiving your revised manuscript.

Kind regards,

Fulvio D'Acquisto, PhD

Academic Editor

PLOS ONE

Journal Requirements:

"The funders had no role in study design, data collection and analysis, decision to

publish, or preparation of the manuscript."

We note that one or more of the authors are employed by a commercial company: "Meta-IQ, ApS"

Please respond by return email with an updated Funding Statement and Competing Interests Statement and we will change the online submission form on your behalf.

Additional Editor Comments (if provided):

Please try to address as many reasonable comments as possible.

Reviewers' comments:

Reviewer's Responses to Questions

**Comments to the Author**

1. Is the manuscript technically sound, and do the data support the conclusions?

Reviewer #1: No

Reviewer #2: Yes

2. Has the statistical analysis been performed appropriately and rigorously? 

Reviewer #1: Yes

Reviewer #2: Yes

3. Have the authors made all data underlying the findings in their manuscript fully available?

Reviewer #1: Yes

Reviewer #2: Yes

4. Is the manuscript presented in an intelligible fashion and written in standard English?

Reviewer #1: Yes

Reviewer #2: Yes

5. Review Comments to the Author

Reviewer #1: Authors of this study have demonstrated that etomoxir effectively mitigates the loss of myelin and other pro-inflammatory, degenerative processes in brain tissue of mouse and rat models of experimental autoimmune encephalitis (EAE), which recapitulates many of the same pathological characteristics as MS in humans. Despite this interesting finding, authors show only superficially descriptive data here and there is no mechanistic insights provided as to why this might be happening. They make the argument that inhibition of CPT1A by etomoxir is a primary reason for the therapeutic effect, yet there are no convincing measurements taken here to support that. Specific areas where this study would be strengthened are as follows:

1) Although etomoxir is indeed an irreversible inhibitor of CPT1A, it also has many off-target effects on other lipid metabolism pathways, including activation of PPARs, and some effects on CoA metabolism. These are only some of the known off-target effects of this molecule, there likely are many more. Thus, many experiments are needed to confirm that CPT1A inhibition is the primary MoA for etomoxir in a study such as this. These experiments are not sufficiently performed in this study. Specifically, no measurements of fatty acid oxidation are performed in any cell or tissue from the EAE models to determine if etomoxir is having an effect on this pathway. The immunostaining for CPT1A protein in the brain slices shown here by the authors is essentially meaningless, as this could be driven by multiple players, including the disease pathology. The functional effect of etomoxir on blunting fatty acid oxidation is not confirmed.

2) It is not at all clear, given the experimental design here, which cell type is relevant to the etomoxir effect. Is CPT1A inhibition more important in oligodendrocytes, lymphocytes, macrophages, or all of them? This is not at all clear from this study. Work in cell culture models might be useful to sort this out, or single-cell experiments from the EAE models.

3) Given the numerous off-target effects of etomoxir, an additional experiment that should be considered is to use a targeted CPT1A-deficient mouse line to confirm that CPT1A is mechanistically involved in the progression of EAE. This would greatly strengthen the etomoxir data.

4) Use of ferritin staining as a marker of ‘mitochondrial dysfunction’ is dubious, at best. If fatty acid oxidation is pathologically involved in EAE, then having some indication of mitochondrial fatty acid oxidation in the tissues involved in this process is essential here. It is well-known that mitochondrial fatty acid oxidation causes more oxidative stress, and in turn this could lead to degeneration of oligodendrocytes and inflammation, but these lines of investigation have not been followed here, and the use of ferritin as a marker does not address these questions in any meaningful way.

5) More discussion about EAE as a model of MS in humans is much needed here. What are the similarities, and what are the limitations?

Reviewer #2: The article entitled,” Pharmacological Inhibition of carnitine palmitoyltransferase 1 inhibits and reverses experimental autoimmune encephalitis in rodents” by Morkholt et al., describes the mechanism of action of a CPT1 inhibitor in vivo. Overall, experiments performed and data analysis are appropriate, and the manuscript can be published in PLOS One after the authors provide some clarification.

1. Fig 3a and b. The authors indicate that rats that received CPT1 inhibitor etomoxir showed lower clinical scores compared to placebo. However, the results were significant only for day 11 data. Furthermore, there was a significant change in body weight. In fact, compared to the change in body weight, the clinical score was less significant. The authors need to explain this.

2. The respiratory chain is an important aspect of cellular metabolism. Inhibition of CPT1 is affected in all cells in the body. Is there any selectivity with CPT1 inhibitors only to neuronal cells?

6. PLOS authors have the option to publish the peer review history of their article (what does this mean?). If published, this will include your full peer review and any attached files.

Reviewer #1: No

Reviewer #2: No

---

## [Author Response · Author response to Decision Letter 0]

11 May 2020

PLOS ONE Decision: Revision required

PONE-D-19-35522

Pharmacological inhibition of carnitine palmitoyl transferase 1 inhibits and reverses experimental autoimmune encephalitis in rodents

Reviewer #1: Authors of this study have demonstrated that etomoxir effectively mitigates the loss of myelin and other pro-inflammatory, degenerative processes in brain tissue of mouse and rat models of experimental autoimmune encephalitis (EAE), which recapitulates many of the same pathological characteristics as MS in humans. Despite this interesting finding, authors show only superficially descriptive data here and there is no mechanistic insights provided as to why this might be happening. They make the argument that inhibition of CPT1A by etomoxir is a primary reason for the therapeutic effect, yet there are no convincing measurements taken here to support that. Specific areas where this study would be strengthened are as follows:

1) Although etomoxir is indeed an irreversible inhibitor of CPT1A, it also has many off-target effects on other lipid metabolism pathways, including activation of PPARs, and some effects on CoA metabolism. These are only some of the known off-target effects of this molecule, there likely are many more. Thus, many experiments are needed to confirm that CPT1A inhibition is the primary MoA for etomoxir in a study such as this. These experiments are not sufficiently performed in this study. Specifically, no measurements of fatty acid oxidation are performed in any cell or tissue from the EAE models to determine if etomoxir is having an effect on this pathway. The immunostaining for CPT1A protein in the brain slices shown here by the authors is essentially meaningless, as this could be driven by multiple players, including the disease pathology. The functional effect of etomoxir on blunting fatty acid oxidation is not confirmed.

Response: Several studies have shown that etomoxir increases the glucose metabolism and decreases the lipid metabolism in both mice and humans[1–5]. Furthermore, we have measured the glucose levels in serum from EAE mice, which is illustrated in the figure below.

Cpt1a EAE (n=4), WT EAE placebo (n=3) and WT EAE etomoxir (n=3).

2) It is not at all clear, given the experimental design here, which cell type is relevant to the etomoxir effect. Is CPT1A inhibition more important in oligodendrocytes, lymphocytes, macrophages, or all of them? This is not at all clear from this study. Work in cell culture models might be useful to sort this out, or single-cell experiments from the EAE models.

Response: CPT1a is expressed in all cells (except in muscle and adipose tissue). We and others have shown in the past that lymphocytes, macrophages dendritic cells etc. are all affected. In addition, astrocytes have shown to have upregulated CPT1a. The theory is that most cells are affected by the drug (etomoxir) and therefore, change the balance in the body to a more glucose metabolism, resulting in the clinical effect on the disease.

3) Given the numerous off-target effects of etomoxir, an additional experiment that should be considered is to use a targeted CPT1A-deficient mouse line to confirm that CPT1A is mechanistically involved in the progression of EAE. This would greatly strengthen the etomoxir data.

Response: These data have in the meantime been published in Scientific Reports where it has been shown that CPT1a knock down mice are resistant to EAE development[6].

We have now referred to this study in section “Discussion” p. 17 l. 361-363.

4) Use of ferritin staining as a marker of ‘mitochondrial dysfunction’ is dubious, at best. If fatty acid oxidation is pathologically involved in EAE, then having some indication of mitochondrial fatty acid oxidation in the tissues involved in this process is essential here. It is well-known that mitochondrial fatty acid oxidation causes more oxidative stress, and in turn this could lead to degeneration of oligodendrocytes and inflammation, but these lines of investigation have not been followed here, and the use of ferritin as a marker does not address these questions in any meaningful way.

Response: Dysfunctional mitochondria and decreased brain metabolism can promote an environment for enhanced oxidative stress. In the manuscript published in meantime, we have demonstrated that EAE severity correlates with the gene expression of proteins involved in oxidative stress such as HO- 1 and NOX2[6]. The results showed that the expression of NOX2 and HO-1was significantly increased in the WT EAE animals. These results were in accordance to the expression of MBP indicating degradation of myelin in the EAE animals.

Ferritin, the iron storage protein, can when reaching excessive levels cause ROS. The levels of ferritin are found to be increased in MS patients compared to normal controls as an attempt to protect against the free levels/excess of iron[7].

We have now made changes to this comment as well as referred to the study in section “Discussion” p. 19 l. 404 and l. 406-409.

5) More discussion about EAE as a model of MS in humans is much needed here. What are the similarities, and what are the limitations?

Response: The experimental autoimmune encephalomyelitis model is an in vivo state-of-art animal model of MS as it mimics the pathological and clinical features present in the disease.

We have included more discussion of the use of EAE as a model for MS in section “Discussion” p. 20 l. 429-435.

Reviewer #2: The article entitled,” Pharmacological Inhibition of carnitine palmitoyltransferase 1 inhibits and reverses experimental autoimmune encephalitis in rodents” by Morkholt et al., describes the mechanism of action of a CPT1 inhibitor in vivo. Overall, experiments performed and data analysis are appropriate, and the manuscript can be published in PLOS One after the authors provide some clarification.

1) Fig 3a and b. The authors indicate that rats that received CPT1 inhibitor etomoxir showed lower clinical scores compared to placebo. However, the results were significant only for day 11 data. Furthermore, there was a significant change in body weight. In fact, compared to the change in body weight, the clinical score was less significant. The authors need to explain this.

Response: This is a good observation and comment of the reviewer. In this rat model the development of EAE was highly aggressive, and all placebo animals died within these 11 days. The effect of the disease induction on the feeding behavior has been found to be preceding the clinical score. In this model weight change is a first indication of disease induction, which therefore has a higher significance.

2) The respiratory chain is an important aspect of cellular metabolism. Inhibition of CPT1 is affected in all cells in the body. Is there any selectivity with CPT1 inhibitors only to neuronal cells?

Response: Same answer as to Reviewer #1 question 2).

CPT1a is expressed in all cells (except in muscle and adipose tissue). We and others have shown in the past that lymphocytes, macrophages dendritic cells etc. are all affected. In addition, astrocytes have shown to have upregulated CPT1a. The theory is that most cells are affected by the drug (etomoxir) and therefore, change the balance in the body to a more glucose metabolism, resulting in a reduced inflammation, reduced oxidative stress and the clinical effect on the disease.

References

1. Timmers S, Nabben M, Bosma M, Bree B Van, Lenaers E, Beurden D Van. Augmenting muscle diacylglycerol and triacylglycerol content by blocking fatty acid oxidation does not impede insulin sensitivity. PNAS. 2012;109: 11711–11716. doi:10.1073/pnas.1206868109

2. Lopaschuk GD, Mcneil GF, Mcveigh JJ. Glucose oxidation is stimulated in reperfused ischemic hearts with the carnitine palmitoyltransferase 1 inhibitor , Etomoxir. 1989; 175–179.

3. Ratheiser K, Schneewiess B, Waldhausl W, Fasching P, Korn A, Nowotny P. Inhibition by Etomoxir of Carnitine Palmitoyltransferase I Reduces Hepatic Glucose Production and Plasma Lipids in Non-Insulin-Dependent Diabetes Mellitus. Metabolism. 1991;40: 1185–1190.

4. Shriver LP, Manchester M. Inhibition of fatty acid metabolism ameliorates disease activity in an animal model of multiple sclerosis. Sci Rep. 2011;1: 6–11. doi:10.1038/srep00079

5. Mørkholt AS, Wiborg O, Nieland JGK, Nielsen S, Nieland JD. Blocking of carnitine palmitoyl transferase 1 potently reduces stress-induced depression in rat highlighting a pivotal role of lipid metabolism. Sci Rep. 2017;7. doi:10.1038/s41598-017-02343-6

6. Mørkholt AS, Trabjerg MS, Krystian M, Oklinski E, Bolther L, Kroese LJ, et al. CPT1A plays a key role in the development and treatment of multiple sclerosis and experimental autoimmune encephalomyelitis. Sci Rep. 2019; 1–11. doi:10.1038/s41598-019-49868-6

7. Williams R, Buchheit C., Berman NE., Levine S. Pathogenic implications of iron accumulation in multiple sclerosis. J Neurochem. 2012;120: 7–25. doi:10.1111/j.1471- 4159.2011.07536.x.Pathogenic

---

## [Editor Report · Decision Letter 1]

28 May 2020

Pharmacological inhibition of carnitine palmitoyl transferase 1 inhibits and reverses experimental autoimmune encephalitis in rodents

PONE-D-19-35522R1

Dear Dr. Nieland,

We are pleased to inform you that your manuscript has been judged scientifically suitable for publication and will be formally accepted for publication once it complies with all outstanding technical requirements.

With kind regards,

Fulvio D'Acquisto, PhD

Academic Editor

PLOS ONE
---

## [Editor Report · Acceptance letter]

1 Jun 2020

PONE-D-19-35522R1 

Pharmacological inhibition of carnitine palmitoyl transferase 1 inhibits and reverses experimental autoimmune encephalitis in rodents 

Dear Dr. Nieland:

I am pleased to inform you that your manuscript has been deemed suitable for publication in PLOS ONE. Congratulations! Your manuscript is now with our production department. 

With kind regards,

on behalf of

Professor Fulvio D'Acquisto 

Academic Editor

PLOS ONE